# Diagnostic and Prognostic Value of Lung Ultrasound Performed by Non-Expert Staff in Patients with Acute Dyspnea

**DOI:** 10.3390/diagnostics15141765

**Published:** 2025-07-13

**Authors:** Greta Barbieri, Chiara Del Carlo, Gennaro D’Angelo, Chiara Deri, Alessandro Cipriano, Paolo De Carlo, Massimo Santini, Lorenzo Ghiadoni

**Affiliations:** 1Emergency Department, University Hospital of Pisa, Via Paradisa 2, 56124 Pisa, Italy; c.delcarlo4@studenti.unipi.it (C.D.C.); gennaro.dangelo@unipi.it (G.D.); c.deri1@studenti.unipi.it (C.D.); pa.decarlo01@gmail.com (P.D.C.); ma.santini@ao-pisa.toscana.it (M.S.); lorenzo.ghiadoni@unipi.it (L.G.); 2Department of Clinical and Experimental Medicine, University of Pisa, 56127 Pisa, Italy

**Keywords:** lung, ultrasound, dyspnea, Emergency Department, emergency medicine

## Abstract

**Background/Objectives**: Dyspnea is one of the main causes of visits to the Emergency Department (ED) and hospitalization, with its differential diagnosis representing a challenge for the clinician. Lung ultrasound (LUS) is a widely used tool in ED. The objective of this study was to evaluate the impact of LUS, performed by a non-expert operator, in determining diagnosis and prognosis of patients with dyspnea. **Methods**: A total of 60 patients presenting with dyspnea at the ED were prospectively enrolled and underwent LUS examination by a medical student, after brief training, within 3 h of triage. LUS findings were classified into four patterns: N.1, absence of notable ultrasound findings, attributable to COPD/ASMA exacerbation; N.2, bilateral interstitial syndrome, suggestive of acute heart failure; N.3, subpleural changes/parenchymal consolidations, suggestive of pneumoniae; and N.4, isolate polygonal triangular consolidation, attributable to infarction in the context of pulmonary thromboembolism. **Results**: The diagnostic hypothesis formulated after LUS was compared with the final diagnosis after further investigations in the ED, showing agreement in 90% of cases. The mean LUS score value was higher in patterns N.2 (18.4 ± 8.5) and N.3 (17 ± 6.6), compared to patterns N.1 and N.4 (9.8± 6.7 and 11.5 ± 2.1). Given the high prevalence of pattern N.2, the diagnostic accuracy of LUS in this context was further evaluated, showing a sensitivity of 82% and specificity of 100%. In terms of the prognostic value of LUS, hospitalized patients had a higher LUS score compared to those discharged (17.3 ± 8.1 vs. 8.5 ± 6.8, *p* value 0.004). A similar trend was obtained in the subgroup of patients requiring non-invasive ventilation (NIV), who present a higher LUS score (21.1 ± 6.6 vs. 13.1 ± 8.1, *p* value 0.002). When considering a combined outcome (death and NIV), patients with worse outcomes more often had a LUS score > 15 (*p* value < 0.001). **Conclusions**: In conclusion, this study confirms that LUS is a very useful tool in the ED, assisting the clinical evaluation for diagnosis, treatment decision, and determination of the appropriate care setting for patients with acute dyspnea. Its short learning curve allows even non-expert staff to use it effectively.

## 1. Introduction

Dyspnea is one of the most frequent symptoms in patients presenting to the Emergency Department (ED), representing a significant diagnostic challenge due to its association with a wide range of pathological conditions, including heart failure, pneumonia, and pulmonary embolism. Timely and accurate diagnosis is crucial to establish appropriate treatment and improve clinical outcomes [1,2].

In this context, lung ultrasound (LUS) is an established rapid, reliable, and non-invasive diagnostic tool, with broad applicability in the emergency setting [3]. Numerous studies highlighted the good sensitivity and specificity of LUS in assessing the aetiologias of dyspnea. Compared to chest X-rays, lung ultrasound offers greater diagnostic accuracy [4,5,6]. In addition, LUS allows for rapid identification of sonographic signs typical of various pathological conditions [7,8,9,10].

A further significant advantage is the possibility of performing the examination directly at the patient’s bedside, reducing diagnostic times, costs, and allowing for more timely therapeutic intervention [4,11].

The repeatability of the investigation also allows continuous monitoring of pulmonary conditions, facilitating the adaptation of therapeutic strategies based on the patient’s clinical evolution [12].

Lung ultrasound demonstrated important utility in the prognostic stratification of patients with dyspnea [13]. The presence and extent of B-lines have been correlated with the severity of heart failure [14] and an increased risk of adverse events [13,15]. Similarly, the detection of pleural effusions or lung consolidations can provide valuable information on the prognosis and efficacy of established therapies [5,16,17].

A significant aspect that has favored the widespread adoption of LUS in emergency departments is its relatively short learning curve. Studies have shown that, through structured training programs, physicians can acquire adequate skills in the performance and interpretation of lung ultrasound in a relatively short time [18,19]. This facilitates the integration of the method into daily clinical practice, improving the management of patients with dyspnea and optimizing the use of healthcare resources [12].

## 2. Materials and Methods

### 2.1. Aim of the Study

The aim of this study was to evaluate the ability of LUS applied with a standardized method by a non-expert operator to perform an adequate diagnostic assessment of a patient with dyspnea in the ED. We also verified whether the ultrasound score (LUS-score) used was able to predict some negative outcomes, such as the need for hospitalization, death, and the need for non-invasive ventilation.

### 2.2. Study Design and Population

A prospective study for dyspnea was conducted on a total of 60 patients with access to the ED of the University Hospital of Pisa between 28 January and 30 April 2024. All included patients underwent LUS at the ED within 3 h of triage, an examination was performed before clinical evaluation, pharmacological treatment, and instrumental tests. The examination was performed by a medical student not involved in the diagnostic-therapeutic decision making of the patient and informed exclusively of the reason for access. Blood and instrumental tests were not examined by the operator. The result of the LUS was not disclosed to the doctors in charge of the patient’s care. Only after the closure of the ED report, the anamnestic, clinical, laboratory, and instrumental data of the patients were collected and analyzed. The outcomes taken into consideration were discharge to home, need for non-invasive mechanical ventilation (NIV) at the ED and during hospitalization, and death.

The research followed the Declaration of Helsinki ethical principles and the international standards of Good Clinical Practice. The Local Ethics Committee approved this protocol on 13 July 2020 (protocol number 17828).

### 2.3. Lung Ultrasound Methodology

The US images were obtained using a convex probe placed in transverse approach, using a depth of approximately 10 cm and focusing on the pleural line. A standardized 16-space scheme was used that allows each hemithorax to be divided into eight areas, widely used in the literature in different clinical contexts [20,21]. Each scanned area was recorded with a clip lasting at least 5 s, aimed at an accurate interpretation of each finding later. Each area was assigned a numerical score based on the degree of lung aeration (LUS score), a methodology widely validated in the literature [19,22]. The scores are listed as follows:Score 0: normal aeration (A lines or less than 3 B lines).Score 1: 3 or more separate B lines or coalescing B lines occupying less than 50 percent of the screen.Score 2: B lines occupying more than 50 percent of the screen.Score 3: lung consolidation.

The final score, obtained by counting all the values obtained in the 16 areas, indicates a decrease in lung aeration as the score increases, with a minimum value of 0 and a maximum of 48. The inferior vena cava (IVC) was also examined at 1.5 cm from its outlet into the right atrium through the subcostal window, considering its maximum and minimum diameter. The examination was performed by a non-expert operator after 15 h of training and the performance of 30 independent examinations. The theoretical training included the basic semiotics of LUS, the main US patterns identification, the B lines counting and the LUS finding associated with the clinical context interpretation. Before starting the enrollment, the student was trained to independently perform a standardized recording of clips for each thoracic area, subsequently reviewed and commented with an expert tutor, to verify the concordance of interpretation. The student began enrolling patients when he had achieved the ability to obtain accurate and well-visualized ultrasound scans and after verifying an interpretative agreement of at least 90% with respect to the expert sonographer’s judgment.

### 2.4. Diagnostic Hypothesis and LUS Patterns

To formulate a diagnostic suspicion based on US examination alone, elements suggestive of 4 typical US patterns described in the literature were identified (Figure 1). In case of evidence of sliding and A lines, without significant pathological findings, the suspicion of an exacerbated respiratory disease such as asthma or COPD was raised (PATTERN N.1). In case of bilateral interstitial syndrome, with multiple diffuse B lines with a gravitational distribution, such as from water overload, we made a presumptive diagnosis of acute cardiac decompensation (PATTERN N.2). In the presence of subpleural alterations or parenchymal consolidations, infectious pneumonia was suspected (PATTERN N.3). In case of a single polygonal consolidation as the only pathological finding, we raised the suspicion of pulmonary infarction in the context of pulmonary thromboembolism (PATTERN N.4).

The cases analyzed were traced back to only one of the 4 patterns described above, with a view to simplification. Subsequently, the concordance between the US diagnostic suspicion and the conclusive diagnosis made in the ED, based on all the clinical, laboratory and instrumental data, was verified.

### 2.5. Statistical Analysis

The data was analyzed with the NCSS statistical program. Data are expressed as mean ± standard deviation, median and interquartile range (IQR) for continuous numeric variables, and as percentage for categorical variables. Differences between groups were analyzed with a parametric test (Student’s *t*-test) for normally distributed variables and a non-parametric test (Mann–Whitney U test) for non-normally distributed variables. χ^2^ test was used for comparisons between variables expressed in the form of frequencies. The *p* value 0.05 was considered statistically significant. To evaluate the impact of the study variables in predicting adverse outcomes, odds ratios were calculated using univariate logistic regression analysis.

## 3. Results

The 60 patients analyzed (M: 31; F: 29) had a mean age of 75.8 ± 12.4 years, ranging from 44 to 90 years. Table 1 shows the main comorbidities found in the population studied.

### 3.1. Diagnostic Value of LUS Score

Figure 2 shows the concordance between the clinical, laboratory, and instrumental diagnosis formulated in ED and the suspicion formulated through the identification of the four LUS patterns. LUS was able to propose a correct diagnosis in 90% of cases, for a total of 54 cases out of 60. Table 2 shows the average value of the total LUS scores, divided by each suspected pathology. It emerges that the highest LUS score is for Pattern N.2 and Pattern N.3. The histogram shows a discrepancy for Pattern 2 in six patients, in whom the main conclusive diagnosis was other than heart failure. In all these six cases, the main diagnosis identified after the ED investigations was pneumonia. However, all these patients had more than two comorbidities, particularly in the cardiovascular field.

Subsequently, we investigated the diagnostic power of the US examination in Pattern 2, obtaining excellent sensitivity and specificity values, as shown in Table 3. In this group, to confirm the diagnosis formulated, the maximum diameter of the IVC was statistically higher (*p* value 0.01) compared to the other groups (Table 4).

### 3.2. LUS Prognostic Value

The mean LUS score value in our sample was 15.5 ± 8.5. In patients older than 79 years, equal to the median of our observations, the mean LUS score was significantly higher than in the subgroup of younger patients, 18.9 ± 7.3 vs. 12.3 ±8.4, *p* value 0.02. A similar result emerged for patients with multiple comorbidities, with mean LUS score values of 18.8 ± 8.0 compared to the remaining population, 11.1 ± 7.1, *p* value 0.002.

The ability of the LUS score to predict negative outcomes was also evaluated. Figure 3 shows that patients requiring hospitalization and deceased patients had a significantly higher LUS score than discharged patients and survivors, with *p*-values of 0.04 and 0.004, respectively. A similar result emerges regarding the need for ventilation in the ED and during hospitalization. Figure 4 shows that patients with respiratory failure requiring NIV had significantly higher LUS score values. In all cases, patients were treated with traditional oxygen therapy with a Venturi mask before the application of NIV, without success. Analysis of oxygen saturation values at admission to the ED showed significantly lower values in the subgroup of patients requiring NIV, 89.4 ± 7.2 vs. 95.1 ± 3.4, *p* value < 0.001.

For the combined outcome of death and need for NIV, a cut-off of 15 was applied (equal to the median of our LUS score observations). Table 5 shows that patients with a LUS score greater than 15 have a significantly higher probability of a negative outcome, with a *p*-value = 0.001. This data is confirmed by univariate logistic analysis, which shows an odds ratio of 8, 2 for the LUS score with a cut-off of 15 (Table 6).

## 4. Discussion

LUS is a useful tool, easy to apply in all care settings [7,12] and has become an indispensable element in the assessment of the patient in the Emergency Department [1,3]. Several studies have demonstrated its usefulness in different phases of hospitalization, starting from the differential diagnosis [5,10,23], to monitoring [3,12,13], to prognosis [15,24,25], and in follow-up [21] in different diseases. Our study aimed to verify the diagnostic and prognostic efficacy of LUS applied by a non-expert operator in the patient presented to the ED with dyspnea.

The studied population were of an older age (75 years), in agreement with data showing respiratory difficulty as the primary cause of medical visits in subjects over 65 years of age [26]. Many of the patients appeared to suffer from multiple comorbidities, of which were arterial hypertension and heart failure. Furthermore, 30% suffered from chronic respiratory system diseases (mostly COPD) and more than half of the total had a positive history of smoking. Therefore, this is a representative sample of the population that accesses the ED daily for reported dyspnea [2].

The LUS examination was performed according to a standardized topographic scheme previously proposed in the literature, applying a numerical scoring for each thoracic area [19,22,27,28]. In particular, a 16-space scheme LUS with scoring has proven to be a reliable tool for early diagnosis of various pathologies, for obtaining objective results, and enabling re-evaluations over time [21]. The LUS findings were grouped into four main patterns associated with the respective presumptive diagnoses, based on what is described in the literature regarding the typical US aspects in different pathological conditions [4,7,8,9,10,17,29].

The standardization of the methodology, useful for objectifying the findings, is a crucial element for the application of the technique by non-expert operators. In our study, the tests were performed by a medical student after completing a short theoretical–practical training.

The innovative element of our study is the use of early US alone in formulating a diagnosis by a student with limited experience, using reference LUS patterns. In our population, it was possible to correctly predict the diagnosis in 90% of the patients studied, confirming the usefulness of the method in the differential diagnosis of dyspnea, as demonstrated in various other contexts [10,11,30]. Those findings were particularly relevant for acute heart failure, considering the high prevalence in our population. We verified the diagnostic power of LUS in this context, observing a sensitivity of 82%, a specificity equal to 100%. Further confirmation of the correctness of the diagnosis is the mean value of the inferior vena cava, significantly higher in the Pattern 2 subgroup than in the rest of the patients, evidence of a state of fluid overload [14,31]. It has to be pointed out that the high level of agreement with other patterns, despite the lower numerosity of diagnosis different from heart failure, cannot support definitive analysis. A careful analysis of the six cases with mismatches showed patients in whom the LUS had suggested the diagnosis of heart failure and instead the final diagnosis of the ED was pneumonia. However, all six patients had multiple cardiovascular comorbidities, a fact that suggests the coexistence of multiple ultrasound patterns, which are often difficult to interpret without an adequate clinical context.

Previous studies have demonstrated the effectiveness of LUS application by inexperienced or non-healthcare operators. A multicenter randomized controlled trial conducted by Baker et al. examined the use of LUS by non-experts in the detection of pulmonary edema. The results indicated that, with adequate training, these operators can achieve good diagnostic accuracy, comparable to that of traditional methods. Furthermore, the implementation of LUS led to reduced costs and improved clinical outcomes, highlighting the importance of structured training programs for non-specialists [32]. Swamy et al. conducted a systematic review to assess the ability of non-physicians to perform and interpret LUS. The review found that, despite some variation in skills, with standardized training, non-physician operators can effectively perform and interpret LUS. This suggests that LUS can be successfully integrated into clinical settings even when performed by non-physician personnel, expanding access to this diagnostic tool [18]. De Molo et al. evaluated the reliability of LUS in this setting, finding good agreement between operators with different levels of experience [33]. During the COVID-19 pandemic, LUS assumed a crucial role in the diagnosis and monitoring of patients [22,23,29].

A second important result of the study regards the association of the LUS score with clinical endpoints, supporting a significant prognostic value of this method. In the context of the COVID-19 pandemic, LUS demonstrated a relevant prognostic role. A study of elderly nursing home residents with COVID-19 found that high lung ultrasound scores were significantly associated with increased mortality [34]. Similar findings emerged through ultrasound assessments performed in the ED [24,35].

Several studies have shown that in patients with heart failure, the presence of a high number of B-lines is associated with a significant increase in the risk of mortality and hospitalizations for heart failure [25,31,36]. In particular, a number greater than 15 B-lines at the time of hospital discharge is significantly associated with an increased risk of all-cause mortality or rehospitalization for heart failure [13,15].

In our sample, the LUS score obtained from examinations in ED was shown to predict some negative outcomes, including significantly higher values in the group of deceased patients as compared to survivors, as well as for a similar trend in hospitalized patients compared to discharged patients and in patients requiring non-invasive ventilation. Those data suggest a crucial role for LUS in the clinical and management decisions in the ED, as well as in the choice of the treatment setting and the ventilatory strategy [20,30].

Finally, when our population was divided based on the combined outcome of death and the need for NIV using the LUS score with a cutoff of 15 (equal to the median value of our observations and already used in numerous other studies [15,37]). The analysis shows that subjects with a LUS score > 15 are more likely to experience early or late negative outcomes (*p* value 0.001). This is confirmed by univariate logistic analysis, in which a LUS score greater than 15 presents an OR of 8.2.

## 5. Conclusions

LUS, applied by a medical student in the ED with a standardized 16-space scheme, numerical score, and identification of specific patterns, was associated with a correct diagnostic suspicion in 90% of the patients examined without taking into consideration any other clinical, laboratory, or imaging elements. A specific LUS pattern had a diagnostic power particularly relevant in recognizing patients with acute heart failure, in association with the evaluation of the inferior vena cava.

Despite the limitations of a small sample size and monocentric evaluation, the total LUS score also demonstrated a significant prognostic value in identifying numerous negative outcomes, among which the need for non-invasive ventilation emerges.

Safety, speed, and manageability make the US a very useful tool in the ED, to be interpreted as a natural complement to the clinical evaluation of the emergency physician, in consideration of its role in the context of rapid decisions about the treatment and care setting of the patient with acute dyspnea.

## Figures and Tables

**Figure 1 diagnostics-15-01765-f001:**
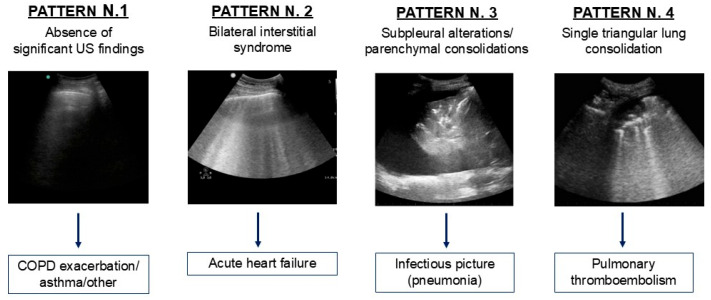
Ultrasound patterns identified in the study population, with respective presumptive diagnoses. US: ultrasound; COPD: chronic obstructive pulmonary disease.

**Figure 2 diagnostics-15-01765-f002:**
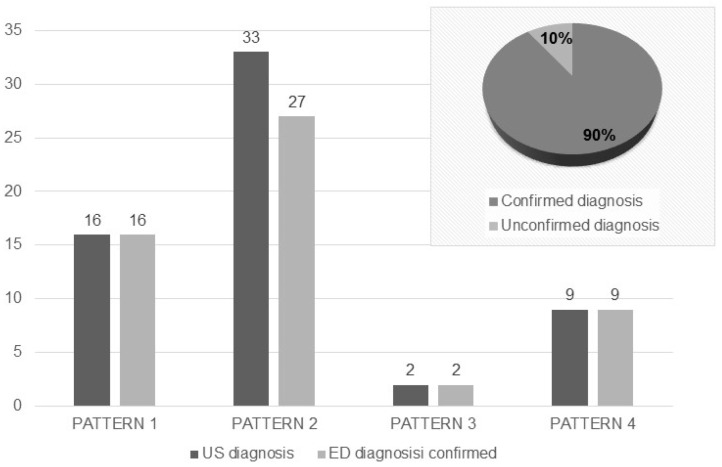
Concordance between the diagnosis advanced based on the US and that formulated at the end of the ED investigations. US: ultrasound; ED: Emergency Department.

**Figure 3 diagnostics-15-01765-f003:**
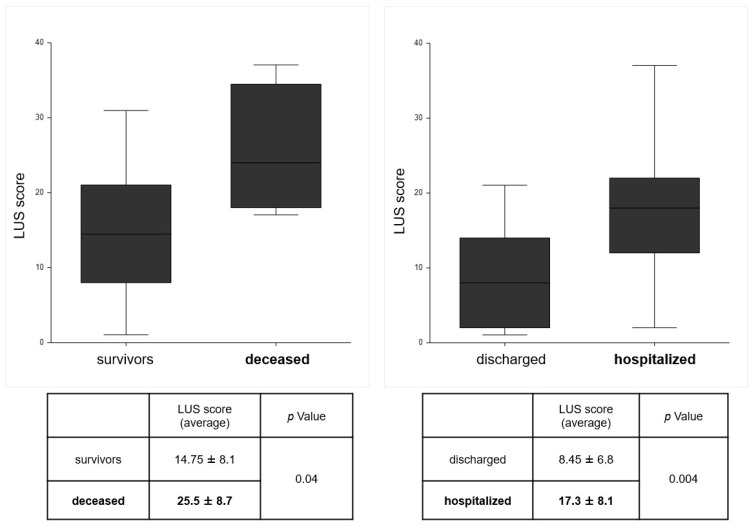
LUS score values in the population divided by mortality and hospitalization. LUS: lung ultrasound.

**Figure 4 diagnostics-15-01765-f004:**
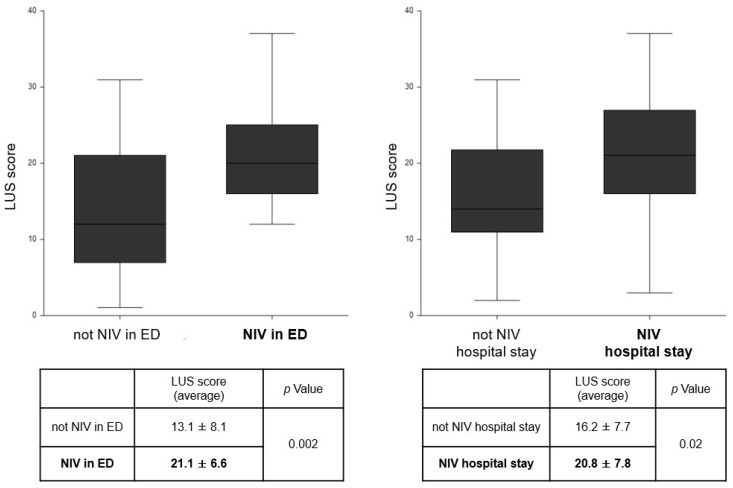
LUS score values in the population divided by non-invasive ventilation requirements. LUS: lung ultrasound; NIV: non-invasive ventilation.

**Table 1 diagnostics-15-01765-t001:** Comorbidity and vital signs on admission in the study population.

	% (*N*)	Age (Mean, Standard Deviation)
Hypertension	60% (36)	79.7 ± 8.3
Heart failure	20% (24)	81.4 ± 5.3
Chronic respiratory disease	30% (18)	77.8 ± 9.7
Diabetes mellitus	33.9% (20)	77.9 ± 9.5
Atrial fibrillation	35% (21)	82.4 ± 4.7
Chronic kidney disease	21.7% (13)	77.5 ± 9.7
Ischemic heart disease	21.7% (13)	79.9 ± 8.8
History of cigarette smoking	51.6% (31)	63.3 ± 14.5
Obesity (BMI > 30)	3.3% (2)	73.5 ± 6.4
Multiple comorbidities (>2)	58.3% (35)	79.7 ± 8.3
	**Mean ± Standard Deviation**	
SaO_2_ (%)	92.9 ± 5.8	
Heart Rate (bpm)	93.6 ± 16.2	
Respiratory Rate (breaths/min)	16.5 ± 5.6	

SaO_2_: Oxygen Saturation.

**Table 2 diagnostics-15-01765-t002:** Mean LUS score values for each US pattern.

Pattern	LUS Score (Mean, Standard Deviation)	Age (Mean, Standard Deviation)
Pattern 1	9.8 ± 6.7	67.25 ± 13.9
Pattern 2	18.4 ± 8.6	80.2 ± 8.5
Pattern 3	17 ± 6.6	78.6 ± 11.3
Pattern 4	11.5 ± 2.1	59.5 ± 20.5

LUS: Lung Ultrasound.

**Table 3 diagnostics-15-01765-t003:** Diagnostic power of US in patients with a conclusive diagnosis of heart failure in the ED.

	ED Diagnosis of Heart Failure (27)	ED Other Diagnoses (33)	*p*-Value
**Pattern 2 (33)**	100% (27)	18.2% (6)	<0.001

ED: Emergency Department. Sensitivity: 100%; Specificity: 82%. Positive predictive value: 82%; Negative predictive value: 100%; Accuracy: 90%.

**Table 4 diagnostics-15-01765-t004:** Mean values of the inferior vena cava in Pattern 2.

	Pattern 2	Pattern 1/3/4	*p*-Value
**Mean diameter of IVC (mm)**	18.6 ± 3.84	15.62 ± 2.3	0.01

IVC: Inferior vena cava.

**Table 5 diagnostics-15-01765-t005:** Predictive power of the LUS score applying a cut-off of 15, using the combined outcome of death and need for ventilation.

	LUS Score < 15 (*n*. 29)	LUS Score > 15 (*n*. 27)	*p* Value
**Positive combined outcome (*n*. 34)**	24 (82.8%)	10 (37.0%)	0.001
**Negative combined outcome (*n*. 22)**	5 (17.2%)	17 (63.0%)	

LUS: Lung Ultrasound.

**Table 6 diagnostics-15-01765-t006:** Univariate logistic regression analysis with LUS score 15 cut-off for combined outcomes of death and non-invasive ventilation.

	ODDS Ratio	CL	Beta	*p* Value
**LUS Score > 15**	8.2	2.4–28.2	2.09	<0.001

LUS: lung ultrasound; CL: confidence limits.

## Data Availability

The original contributions presented in this study are included in the article. Further inquiries can be directed to the corresponding author.

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
