# Peer review of "Diagnostic and Prognostic Value of Lung Ultrasound Performed by Non-Expert Staff in Patients with Acute Dyspnea"

_diagnostics, 2025, doi:10.3390/diagnostics15141765_

Round 1
Reviewer 1 Report
Comments and Suggestions for Authors
In this study, the authors aimed to evaluate the impact of lung ultrasound, performed by a non-expert operator, in determining diagnosis and prognosis of patients with dyspnea.
The choice to use a medical student with minimal training to perform the LUS adds novelty and clinical significance.
The article has well-defined methodology and good statistics.
The tables and figures are clear and correct.
As you mentioned in the study limitations, the sample is small (n=60) and single-center, which limits generalizability. Further multi-center studies with larger populations are needed to validate findings.
The student underwent 15 hours of training and 30 supervised scans, but more specifics on the curriculum and quality control would improve reproducibility.
The article presents 38 references being up to date.
Author Response
Reviewer #1:
Comment 1. In this study, the authors aimed to evaluate the impact of lung ultrasound, performed by a non-expert operator, in determining diagnosis and prognosis of patients with dyspnea.
The choice to use a medical student with minimal training to perform the LUS adds novelty and clinical significance.
The article has well-defined methodology and good statistics.
The tables and figures are clear and correct.
As you mentioned in the study limitations, the sample is small (n=60) and single-center, which limits generalizability. Further multi-center studies with larger populations are needed to validate findings.
Response 1.We thank the reviewer for the comments.
Comment 2.The student underwent 15 hours of training and 30 supervised scans, but more specifics on the curriculum and quality control would improve reproducibility.
Response 2.We thank the reviewer for the opportunity to integrate the methodological section, which is crucial in our work. The details about the student's training have been inserted on page 3, lines 105-113.
Comment 3.The article presents 38 references being up to date.
Response 3.We thank the reviewer for the comments.
Reviewer 2 Report
Comments and Suggestions for Authors
I appriciate the opportunity to review such an interesting manuscript!
The major strengths are the well-provided reseach with a students. Nowadays, US inserted to all medical specialties and highly important for young doctors. The examination was performed by a non-expert operator after 15 hours of training and the performance of 30 independent examinations.
The major limitations are the lack of clinical data of the patients.
My suggestions are as follows:
1. Some information appears in Discussion section is absent in Result.
2. Its not a secret that more severe patients could have higher LUS. Was there any correlation between LUS and patients with a history of more than 2 diseases or with advanced age?
3. The indications for NIV are well knowen. There is no information about the level of saturation at admission, oxygen usage through facial mask or nasal cannula before or instead of NIV.
4. It should be data about age of respondents and analysis of age-dependent pathology.
5. During the COVID-19 pandemic, LUS assumed a crucial role in the diagnosis and monitoring of patients.
It is important to describe the process of research starting with the time of examination (immediately after admission, before electrocardiogram or after, how much did it take…)
6. Several studies have shown that in patients with heart failure, the presence of a high number of B-lines is associated with a significant increase in the risk of mortality and hospitalizations for heart failure. Author’s data suggests a crucial role for LUS in the clinical and management decisions in the ED, as well as in the choice of the treatment setting and the ventilatory strategy.
Could authors provide some algorithm or decision-making tree with LUS and other factors of assessed patients?
7. The analysis shows that subjects with a LUS score > 15 are more likely to experience early or late negative outcomes.
Is this cutoff suitable for all patients? Did the authors check this cutoff point in subgroups of heart-patients and lung disease-patients separately?
Author Response
Reviewer #2:
Comment 1.Some information appears in Discussion section is absent in Result.
Response 1.We verified the correspondence between the data reported in the results and those commented in the discussion, to eliminate discrepancies.
Comment 2.It’s not a secret that more severe patients could have higher LUS. Was there any correlation between LUS and patients with a history of more than 2 diseases or with advanced age?
Response 2.We thank the reviewer for giving us the opportunity integrate our analysis. We found that in our population, patients with two or more comorbidities were more than 58%. The LUS score was significantly higher in elderly patients and those with more than 2 comorbidities, reaching statistical significance. This result suggests that these categories more frequently present pathological conditions causing extensive lung involvement, visible by US. The requested analyses have been included in the results section, page 4, lines 158-162 and table 1.
Comment 3.The indications for NIV are well knowen. There is no information about the level of saturation at admission, oxygen usage through facial mask or nasal cannula before or instead of NIV.
Response 3.We thank the reviewer for this clinically relevant comment, although the indications for NIV are beyond the scope of our study. However, we implemented the clinical information of the population by also including the Oxygen Saturation at admission and verified that the mean values ​​of patients requiring NIV were significantly lower, with p value < 0.001. All patients were subjected to traditional oxygen therapy before starting ventilation, without clinical improvement. The results of these additional analyses have been included in the revised version of the manuscript, in the results section, page 4, lines 168-172 and table 1.
Comment 4.It should be data about age of respondents and analysis of age-dependent pathology.
Response 4.We thank the reviewer for suggestions. Demographic information is available in the Results section on page 3, lines 141-142. We also calculated the mean age of patients based on underlying diseases and in the 4 subgroups identified by US patterns. Finally, we calculated the mean LUS score values ​​in the population divided by age, highlighting significantly higher LUS score ​​in older subjects. Data from additional analyses performed are shown in the Results section, page 4, lines 158-160 and in Tables 1 and 2.
Comment 5.During the COVID-19 pandemic, LUS assumed a crucial role in the diagnosis and monitoring of patients.
It is important to describe the process of research starting with the time of examination (immediately after admission, before electrocardiogram or after, how much did it take…)
Response 5.Thanks to the reviewer for pointing this out. We integrated the information about the timing and methods of the LUS examination, described in the Methods section, page 2, lines 73-79.
Comment 6.Several studies have shown that in patients with heart failure, the presence of a high number of B-lines is associated with a significant increase in the risk of mortality and hospitalizations for heart failure. Author’s data suggests a crucial role for LUS in the clinical and management decisions in the ED, as well as in the choice of the treatment setting and the ventilatory strategy.
Could authors provide some algorithm or decision-making tree with LUS and other factors of assessed patients?
Response 6.We agree with the reviewer's considerations, because US should be included in the decision-making process of the Emergency Department physician, to speed up the patient's diagnostic-therapeutic process. However, the development of a management algorithm for patients with dyspnea is beyond the scope of our study, which mainly aims to demonstrate the diagnostic accuracy of LUS applied with a standardized technique by a non-expert operator.
Comment 7.The analysis shows that subjects with a LUS score > 15 are more likely to experience early or late negative outcomes.
Is this cutoff suitable for all patients? Did the authors check this cutoff point in subgroups of heart-patients and lung disease-patients separately?
Response 7.We thank the reviewer for the comments. We verified the predictive power of the LUS score with a cut-off of 15 also for pathology subgroups, but these analyses were not included in the maniscripst because the subdivision determines a reduction in the sample size, responsible for the lack of statistical significance. Further studies with larger numbers will be needed to verify the applicability of the method in patients with different diseases. However, as described in Discussion on page 12, lines 309-311, a similar score was used in other studies in the literature, demonstrating a comparable predictive capacity.
Reviewer 3 Report
Comments and Suggestions for Authors
- The authors should provide more detailed demographic features of all patients involved in this study.
- Tables and figures should be placed where they are first mentioned in the text.
- For pattern 3, the sample size was too small. Moreover, Pattern 2, which had the largest sample size, showed the lowest accuracy. How do the authors explain this discrepancy?
- The authors should report the final diagnoses of the inconsistent cases in Pattern 2 and analyze the possible underlying reasons.
- The authors have shown the predictive abilities of LUS for patient survival and hospitalization. More importantly, the authors should provide survival difference between patients with the same disease, but with or without LUS examinations, ideally using retrospective data.
Author Response
Reviewer #3:
Comments 1.The authors should provide more detailed demographic features of all patients involved in this study.
Response 1. We thank the reviewer for suggestions. Demographic information is available in the Results section on page 3, lines 141-142. Furthermore, to better enhance the anagraphic aspect, we calculated the mean age of patients based on diseases and in the 4 subgroups identified by US patterns. Finally, we calculated the mean LUS score values ​​in the population divided by age. Data from additional analyses performed are shown in the Results section, page 4, lines 158-160 and in Tables 1 and 2.
Comments 2.Tables and figures should be placed where they are first mentioned in the text.
Response 2. We have followed a template provided by the journal. We are available to modify the layout of figures and tables in the next phase of revision, following the indications of the editorial office.
Comments 3-4 .For pattern 3, the sample size was too small. Moreover, Pattern 2, which had the largest sample size, showed the lowest accuracy. How do the authors explain this discrepancy?
The authors should report the final diagnoses of the inconsistent cases in Pattern 2 and analyze the possible underlying reasons.
Response 3-4. We thank the reviewer for the comments. The discrepancy concerns 6 patients for whom the US examination would have suggested the diagnosis of heart failure (Pattern 2), not confirmed by the final diagnosis of the ED, which had been pneumonia for all 6 cases. By verifying the basic characteristics, a population with more than 2 comorbidities in the cardiovascular area emerged. This data suggests that the thoracic picture could be characterized by "mixed" elements, expression of multiple pathological conditions. Furthermore, we believe that this small divergence is due to the inevitable simplification implemented using 4 distinct US patterns. We integrated this data into the Results section, Page 4, lines 149-152. Additionally, we commented these data in Discussion, page 11, lines 285-290.
Comments 5.The authors have shown the predictive abilities of LUS for patient survival and hospitalization. More importantly, the authors should provide survival difference between patients with the same disease, but with or without LUS examinations, ideally using retrospective data.
Response 5. We agree with the reviewer's considerations, as we believe that the contribution of LUS is crucial in the management of patients with dyspnea. However, it is not the aim of our study to verify the prognostic impact of patients in which LUS has been applied compared to conventional iter. The main objective of our work was to demonstrate the possibility of obtaining valuable information from an examination performed by a non-expert operator, if supported by a good methodology.
Round 2
Reviewer 2 Report
Comments and Suggestions for Authors
The manuscript was corrected well
Reviewer 3 Report
Comments and Suggestions for Authors
The authors have addressed or revised all the issues. Publication is recommended.